# Online Consumer Survey Comparing Different Front-of-Pack Labels in Greece

**DOI:** 10.3390/nu14010046

**Published:** 2021-12-23

**Authors:** Lamprini Kontopoulou, George Karpetas, Εvangelos C. Fradelos, Ioanna V. Papathanasiou, Foteini Malli, Dimitrios Papagiannis, Dimitrios Mantzaris, Morgane Fialon, Chantal Julia, Konstantinos I. Gourgoulianis

**Affiliations:** 1Nursing Department, University of Thessaly, 41500 Larissa, Greece; evagelosfradelos@gmail.com (Ε.C.F.); iopapathanasiou@yahoo.gr (I.V.P.); mallifoteini@yahoo.gr (F.M.); dmantzar@uth.gr (D.M.); 2Medicine Department, University of Thessaly, 41110 Larissa, Greece; kgourg@uth.gr; 3Public Health & Vaccines Lab, Department of Nursing-School of Health Sciences, University of Thessaly, 41110 Larissa, Greece; dpapajon@gmail.com; 4Nutritional Epidemiology Research Team (EREN), Sorbonne Paris Cité Epidemiology and Statistics Research Center (CRESS), Inserm U1153, Inrae U1125, Cnam, Université Sorbonne Paris Nord University, 93017 Bobigny, France; m.fialon@eren.smbh.univ-paris13.fr (M.F.); c.julia@eren.smbh.univ-paris13.fr (C.J.); 5Public Health Department, Avicenne Hospital, Assistance Publique des Hôpitaux de Paris (AP-HP), 93000 Bobigny, France

**Keywords:** nutritional labelling, food policies, front-of-pack nutrition label, promote health, Greek consumers

## Abstract

According to the WHO, front-of-pack nutrition labeling provides simplified nutrition information in the form of symbols, colors or words that can help consumers understand the nutritional quality of food, thus leading them to healthier food choices. It is considered of the utmost importance to explore the knowledge and understanding of consumers about this form of nutrition labeling. The aim of this study was to investigate the understanding and perceptions of Greek consumers in response to five different front-of-pack nutrition labels (FoPLs): the Multiple Traffic Lights, Health Star Rating System, Guideline Daily Amounts, Warning Symbols and Nutri-Score. From April 2021 to June 2021, 1278 participants from Greece took part in an online survey where they were asked to rank three products according to their nutritional quality. The classification process was performed first without FoPLs and then with FoPLs. The ability to classify products according to their nutritional quality was evaluated with multinomial logistic regression models, and the Nutri-Score label presented greater improvements when compared to the GDA label for Greek consumers. The Nutri-Score seemed to better help the Greek consumers rank foods according to their nutritional value.

## 1. Introduction

In Greece, the adult obesity rate, which stands at 17%, is higher than in other southern European countries [1]. The number of obese people in Europe is estimated to have tripled compared to 20 years ago [2]. More specifically for childhood obesity, data from the Childhood Obesity Surveillance Initiative (COSI) study showed that the highest rates of childhood overweight and obesity were observed in Mediterranean countries, such as Cyprus, Greece, Italy and Spain, where more than 40% of boys and girls were overweight, and 19% to 24% of boys and 14% to 19% of girls were obese [3]. In the last two decades, obesity has increased worldwide and is associated with chronic diseases that can adversely affect both the health of individuals and the viability of the health system. Overweight and obesity have also been linked to chronic health conditions, such as diabetes, hypertension, heart disease, stroke, chronic respiratory disease and certain cancers [4,5,6].

Non-communicable diseases (NCDs) threaten the 2030 Agenda for Sustainable Development, as one of the goals of the agenda is to reduce premature deaths from NCDs by one-third by 2030 [7]. Over 40% of deaths in Greece can be attributed to behavioral risk factors [1].

Unhealthy diets are considered to be the leading cause of death and disability worldwide [8,9]. In recent decades, there has been an increase in the consumption of energy-dense, high-fat and high-sugar foods [10]. There has also been an increase in portion sizes [2]; both contribute to the increase in obesity prevalence.

As a result, one of the most important challenges for all countries is the promotion of healthier diets.

One of the ways to intervene and promote health is to use nutrition labeling that may help people to make healthier food choices.

According to the WHO (World Health Organization/European Region), front-of-pack nutrition labels (FoPLs): (a) can provide simplified nutrition information in the form of symbols, colors or words and (b) are considered an economical and cost-effective measure to help consumers understand the nutritional quality of food, thus leading them to healthier food choices [9,11].

FoPLs can be classified into three categories: (a) numerical information according to the nutrient content, (b) interpretive labels and (c) logos with health claims [12,13]. More particularly, FoPL schemes have existed in Europe for decades. In the 1980s, the “Green Keyhole” was adopted in Sweden and Denmark, while in the 2000s, the “Choices” and the “Multiple Traffic Lights” were adopted in the Netherlands and in the United Kingdom, respectively. In 2014, Australia and New Zealand adopted the “Health Star Rating System”, while in 2016, Chile adopted the “Warning Symbols” for each nutrient whose food content is considered high. In parallel with the schemes adopted by government agencies, in 2006 Food Drink Europe, representing the private sector, developed the Guideline Daily Amounts (GDA) scheme, which was later renamed Reference Intakes (RI). The Nutri-Score frontal nutrition declaration scheme was developed in France and officially adopted in 2017 [14].

Discussions are underway in Greece regarding which FoPL scheme is the most suitable for helping Greek consumers understand the nutritional quality of foods at the time of purchase and to orient them towards healthier choices. However, according to a press release, the Hellenic Ministry of Rural Development and Food is quite cautious about the Nutri-Score scheme and proposes a monochrome and descriptive system (not interpretative of the Nutri-Score), as well as the exclusion of single-ingredient products (such as olive oil and honey) and some other products (such as products of designation of origin, products of geographical indication and guaranteed traditional specific products) [15].

For this reason, a study to explore the perception and understanding of different FoPLs by consumers living in Greece, including the Nutri-Score, is considered imperative. According to Pauline Ducrot et al., understanding can be assessed by either subjective or objective criteria. Objective understanding allows for a more accurate measurement than subjective understanding, which can lead to overestimation. More specifically, subjective understanding indicates the degree to which consumers believe they understand a label, while objective understanding can determine whether the information that consumers understand is consistent with the information provided by the label [16]. In addition, according to Grunert et al. [17,18], objective understanding examines and evaluates the accuracy of the information perceived by the consumer concerning the real meaning that suppliers want to communicate. In addition, in the EU Joint Research Center’s recent report on front-of-pack labelling, it is emphasized that short and simple labels achieve the best objective understanding [19]. In Greece, so far, no study has been carried out to investigate the main forms of labeling, including the Nutri-Score to compare the effectiveness of different FoPLs.

The purpose of this study was to investigate the choices, objective understandings and perceptions of Greek consumers in response to five different FoPL systems using a questionnaire in electronic form. Both the questionnaire and the methodology were based on the FOP-ICE (Front-Of-Pack International Comparative Experimental) study that has been conducted thus far in 18 countries [20] and was built on previous research, investigating both the perceptions [21,22] and objective understandings in various populations [16,23]. The following FoPLs were included in the study: the Multiple Traffic Lights (MTL), Health Star Rating System (HSR), Nutri-Score, Warning Symbols and Reference Intakes (RIs), and the foods that were examined, according to their availability in the Greek market, were the following: cakes, pizzas and breakfast cereals.

## 2. Materials and Methods

Participation in this survey was voluntary and anonymous. The questionnaire was officially requested by the French scientific team, and it was provided in English. It was then translated into Greek, initially, by a person who knew fluent English, and then translated from Greek again into English by a different person who also knew fluent English to check for any differences between the texts. The participants expressed their consent as on the first page of the electronic form their consent was requested, and there was an assurance for the observance of the anonymity and confidentiality of the answers. The study was approved by the Ethics Committee of the University of Thessaly (approval report: 24/15 April 2021).

### 2.1. Participants

Participants were recruited by the researchers via social networking, personal mailand professional mail databases (University of Thessaly, Larissa’s Medical Association, Pharmaceutical Association of Larissa, Secondary Education, Larissa Primary Education Parents Association, Second Chance School, Hellenic Air Force, Airport of Volos), and the electronic questionnaire was promoted using the avalanche method.

From the total number of 1402 participants, 124 did not answer the income question and were excluded from the survey. The final number of participants was 1278. Data collection was conducted from April 2021 to June 2021.

Quotas were used based on gender (49.0% male, 51.0% female) and age, which was categorized as follows: from 18 to 30 years old, from 31 to 50 years old and over 51 years old.

Additionally, there was a categorization based on household income level and number of people in the household. According to the 2020 Household Income and Living Conditions Survey report [24] for a typical household income with two economically active people, the classification in the present study was calculated as follows: low <12,320 euros/year, medium 12,334–24,640 euros/year and high >24,640 euros/year). Participants who stated that they did not buy at least two of the three types of food (cakes, cereals and pizza) that they were asked to evaluate in the questionnaire were considered ineligible.

Participants were asked to complete the online survey questionnaire in Greek. In the first part of the survey, participants were asked to answer general questions about gender, age, monthly household income, number of family members, educational level, participation in supermarket shopping and quality of their diet. Moreover, they were asked to self-rate their diet quality, their nutritional knowledge and if they noticed the label during the survey. At the end of the questionnaire, participants were asked to respond to nine questions (formed as statements) related to their perceptions about the label they were assigned to, including awareness, trustworthiness, liking and perceived cognitive workload. The choice of answers was according to a 5-point Likert’s scale (1 = strongly disagree, 5 = strongly agree).

### 2.2. Procedure & Front-of-Pack Nutrition Labels

Participants were asked to respond about three categories of food that are widely available in the Greek market: cakes, pizzas and breakfast cereals. A fictional food brand was used with name “Stofer” to avoid bias among participants [20]. Initially, for each food category, participants were asked to rate a set of three unlabeled products by choosing one of three options for each product:Higher nutritional valueAverage nutritional valueLower nutritional value

There was also the option “I do not know”.

Participants were randomized into one of five groups of FoPLs (RIs, HSR, MTL, Warning Symbols and Nutri-Score) with balanced repartition, resulting in the assignment of approximately 255 participants per group. The participants were then asked to classify the same products with one of the FoPLs in which they were randomized. Products with the five FoPLs labels are shown in Figure 1.

### 2.3. Data Analysis

All analyses were performed in SPSS version 26, IBM (SPSS Inc., Chicago, IL, USA). The statistical significance level was set at α = 0.05.

#### 2.3.1. Food Choice

For each food category, a score between 1 and 3 points was attributed with 1 point for the lowest nutritional quality product, 2 for median nutritional quality product and 3 points for the highest nutritional quality product. Participants who selected “I wouldn’t buy any of these products” were excluded from the analysis. Then, the difference scores between the no label and label condition were calculated for each food category, ranging between −2 and +2. A total score was computed by summing the difference scores of the three food categories, obtaining values between −6 and +6 points. Then, for each FoPL group by food category, the percentage of participants whose food choices worsened or improved between the unlabeled and the FoPL conditions was calculated.

With the purpose of assessing possible relationships between the FoPL type and scores from food choice tasks for all food categories as well as separately for each food category, multinomial logistic regression was used, setting the GDA label as the reference category. Four models were produced; each adjusted for variables of personal characteristics (sex, age, household monthly income level, education level, involvement in grocery shopping, nutrition knowledge, self-reported diet quality and whether the FoPL was noticed during participation in the study).

#### 2.3.2. Objective Understanding

Consumers’ objective understanding of the FoPL was measured by the ability of participants to correctly sort the products in each set according to nutritional quality. Participants who ranked all three products correctly were assigned a score of +1, while participants who made at least one mistake in ranking were assigned a score of −1. A total score for the difference in ranking performance between the two label conditions was calculated for each food category (values in the range of −2 to +2) and for all food categories (ranging between −6 and +6). Percentages of correct ranking in both label conditions were calculated per food category and FoPL type.

With the purpose of assessing possible relationships between ranking performance and FoPL type, multinomial logistic regressions were produced, for all food categories and separately for each food category, setting the GDA label as the reference category. All regression models were adjusted for sex, age, household monthly income level, education level, involvement in grocery shopping, nutrition knowledge, self-reported diet quality and whether the FOPL was noticed during participation in the study.

#### 2.3.3. Perception

To assess participants’ perceptions of food labels, questions were asked (related to perception), and the answers were selected according to a 5-point Likert’s scale (1 = strongly disagree, 5 = strongly agree). Mean values and standard deviations were calculated for each question. Principal component analysis was utilized to explore the dimensionality of the perceptions questionnaire, as well as the contribution of each statement to the respective dimension. Dimensions were produced as the linear combination of statement scores. The total variance explained by each dimension was calculated based on their eigenvalues, and the number of dimensions was selected according to the aggregated percentage of explained variance.

## 3. Results

### 3.1. Characteristics of the Population

The main characteristics of the study population, including sociodemographic data, lifestyle and nutrition-related characteristics, are depicted in Table 1. The sample included 1278 Greek participants, of whom 51% were women, 34.8% were individuals between 31–50 years old, 36.3% declared the median income and 20.7% had a university postgraduate degree. In addition, 65.9% declared that were responsible or co-responsible for grocery shopping, 81.1% reported that they follow a healthy diet and 39.5% had no nutritional knowledge or little nutrition knowledge. In the end, 58.6% reported that they did not or sometimes read the nutritional statement on the back of the product packaging, and 53.4% of participants referenced that they saw the FOP label during the survey.

The percentages of participants per label are: MTL 19.9%, GDA 19.9%, Warning Symbols 20.3%, Nutri-Score 19.9% and HSR 19.9% (Table 2).

### 3.2. Food Choices

A ratio up to 0.5% of participants (dependent on the food category and FoPL) did not select any product and were omitted from the analysis. Another 57 to 69.4% of participants (depending on the food category and FoPL type) did not change their choice between the no label and the FoPL conditions. Compared to the no label condition, the food choice differed significantly in the FoPL condition for all three food categories: pizza, cake and breakfast cereals (McNemar–Bowker test *p*-value = 0.001 for each food category).

The percentage of participants who improved or deteriorated their food choices depending on the product category and its FoPL are shown in Figure 2 for all food categories: pizzas, breakfast cereals, cakes and all FoPLs (MTL, HSR, Nutri-Score, Warning Symbols and RIs). Between 15.7% and 44.7% of participants (depending on the label and the food category) demonstrated an improvement in the nutritional quality of their choices, while between 2% and 28.6% of participants demonstrated a deterioration. More specifically, the Nutri-Score label demonstrated the highest improvement rates (between 25.6% and 44.7% depending on the food category). Following, the GDA demonstrated the second highest improvement rates (between 17.2% and 30.3% depending on the food category). The smallest improvement rates were demonstrated by the Warning Symbols labels (between 18.6% and 22%) and the HSR (between 16.9% and 22.3%). For the MTL, the improvement rates were between 15.7% and 23.5%, yet the deterioration rates were higher than the improvement rates for breakfast cereals (20.4%) and cakes (23.2%). The same is observed only for the HSR in the cakes’ food category (deterioration 28.6%, improvement 18.2%).

For all food categories and FoP labels, except the Nutri-Score, slight to moderate deteriorations were observed compared to the GDA label. For the MTL, statistically significant deteriorations compared to the GDA were observed for all categories combined (OR = 0.71 [0.63–0.79] (*p* < 0.001)), cakes (OR = 0.55 [0.46–0.68] (*p* < 0.001)) and breakfast cereals (OR = 0.59 [0.47–0.76] (*p* < 0.001)). For the Warning Symbols labels, statistically significant deteriorations compared to the GDA were observed for all categories combined (OR = 0.85 [0.77–0.96] (*p* < 0.001)), pizza (OR = 0.67 [0.52–0.88] (*p* < 0.001)) and cakes (OR = 0.79 [0.66–0.96] (*p* < 0.05)). For the HSR, statistically significant deteriorations compared to the GDA were observed for all categories combined (OR = 0.75 [0.67–0.85] (*p* < 0.001)), pizza (OR = 0.54 [0.41–0.72] (*p* < 0.001)) and cakes (OR = 0.54 [0.45–0.66] (*p* < 0.05)). Overall, no significant associations between the Nutri-Score and change in the nutritional quality of the food choices were found compared to the GDA label, in any of the three food categories (Table 3). The low performance of the MTL compared to the GDA may be attributed to the fact that only 16.9% of participants in the MTL group were responsible for grocery shopping, a significantly lower percentage compared to 33.3–37.5% within the other label groups (χ^2^(8) = 52.888, *p* < 0.01). In this frame, not being responsible for grocery shopping was a significant predictor only for the MTL label in the food categories of pizza (OR = 1.62 [1.07–2.47] (*p* < 0.05)), cake (OR = 1.63 [1.07–2.51] (*p* < 0.05)) and breakfast cereals (OR = 1.57 [1.03–2.40] (*p* < 0.05)).

### 3.3. Objective Understanding

The percentages of correct answers for product ranking in the no label and label conditions by FoPL are presented in Figure 3. Participants who selected “I don’t know”, hence not providing the ranking information, were excluded from the analysis. All labels demonstrated increased rates of correct answers compared to the no label condition (between 13.7% and 58% depending on the label and food category). The Nutri-Score label demonstrated the highest improvement rates in the ranking task (between 42.4% and 58% depending on the food category). Following, the HSR demonstrated the second highest improvement rates (between 37.4% and 53.8% depending on the food category). The smallest improvement rates were demonstrated by the GDA (between 13.7% and 44.6%).

In general, all labels present improvements in the ability of participants to correctly rank products according to their nutritional quality in all three food categories, compared to the GDA label. The Nutri-Score label presented the greatest improvements compared to the GDA label (odds ratio (OR): all food categories = 1.36 [1.23–1.51] (*p* < 0.001); pizza = 1.69 [1.39–2.06] (*p* < 0.05); cakes = 1.53 [1.29–1.84] (*p* < 0.001); breakfast cereals = 1.27 [1.05–1.53] (*p* < 0.05)). The MTL also significantly improved the ranking ability of the participants compared to the GDA in all food categories combined (OR = 1.17 [1.06–1.30] (*p* < 0.001)), pizza (OR = 1.29 [1.05–1.90] (*p* < 0.05)) and the cake category (OR = 1.49 [1.24–1.80] (*p* < 0.001)). The HSR also significantly improved the ranking ability of the participants compared to the GDA in all food categories combined (OR = 1.23 [1.11–1.37] (*p* < 0.001)), pizza (OR = 1.34 [1.10–1.64] (*p* < 0.01)) and the cake category (OR = 1.53 [1.27–1.85] (*p* < 0.001)), whereas the Warning Symbols were associated with significant improvements in the ability to correctly rank products when compared to the GDA label in the cake category only (OR = 1.37 [1.14–1.66] (*p* < 0.01)) (Table 4). The Nutri-Score was the label with the highest performance for all food categories, followed by the MTL and HSR. (Table 4).

### 3.4. Perception

The perceptions of consumers were generally favorable for all FoPLs with no significant differences between the different FoPL types (Figure 4). For the item ‘This label is confusing’, the Warning Symbols labels (M = 2.07, SD = 1.07) and the HSR (M = 2.01, SD = 1.09) presented the highest ratings. For the item ‘It should be compulsory for this label to be shown on packaged food products’, the GDA presented the highest rating (M = 4.56, SD = 0.92), while for the item ‘I like this label’, the Nutri-Score was most favored (M = 4.2, SD = 1.04). For the item ‘This label does not stand out’, the Nutri-Score had the lowest rating (M = 1.82, SD = 1.12), and for the item ‘This label is easy to understand’, the Nutri-Score had the highest rating (M = 4.05, SD = 1.12). Moreover, the MTL (M = 2.33, SD = 1.15) and the GDA (M = 2.33, SD = 1.17) presented the highest ratings for the item ‘This label took too long to understand’ and for the item ‘This label provides me with the information I need’ (MTL: M = 3.87, SD = 1.13, GDA: M = 3.76, SD = 1.09). The Nutri-Score (M = 3.76, SD = 1.06) had the highest rating, and the HSR (M = 3.36, SD = 1.11) the lowest rating for the item ‘I trust this label’.

The principal component analysis produced two main dimensions explaining 40.5% and 16% of the total variance, respectively. Table 5 presents the eigenvectors of the various statements on both dimensions. In the first dimension, items such as ‘It should be compulsory for this label to be shown on packaged food products’, ‘I like this label’, ‘This label is easy to understand’, ‘This label provides me with the information I need’ and ‘I trust this label’ present positive loadings, indicating a strong positive correlation with this dimension. On the other hand, items such as ‘This label is confusing’, ‘This label does not stand out’ and ‘This label took too long to understand’ presented higher positive loadings with the second dimension (Table 5).

Based on the results of the principal component analysis, a map positioning each FoPL against the two dimensions was created (Figure 5). The range of the positioning on the first dimension was between −0.5 and 0.8, while the second dimension was between −0.2 and 0.3. Hence, relatively larger differences were observed between labels in the first dimension compared to the second dimension. More specifically, the Nutri-Score and GDA have positive positionings in the first dimension in opposition to the Warning Symbols, HSR and MTL. The Nutri-Score was considered the most trusted, useful and easy to understand FoPL (with the highest positioning in the first dimension). The GDA and MTL were considered time-consuming to understand (with higher positioning in the second dimension). The HSR, with negative positioning mainly in the first dimension, was perceived as less trusted, while Warning Symbols, with negative positionings in both dimensions, were perceived mainly as a confusing label.

## 4. Discussion

This is the first study conducted in Greece comparing the performances of different FoP labels (among them the Νutri-Score). In this online study, the results showed the superiority of the Nutri-Score over the other labels included in our survey, such as the GDA, HSR, MTL & Warning Symbols. The most important differences were observed in terms of the result of objective understanding. The Nutri-Score was the FoPL with the largest increase in the participants’ abilities to correctly classify the nutritional quality of products (between 42.4 and 58% depending on food category), followed by the HSR, which demonstrated the second highest improvement rates (between 37.4% and 53.8% depending on food category). The GDA recorded the smallest improvement rates (between 13.7% and 44.6%). Studies in Mexico and Chile showed that the GDA label is ineffective for choosing the right and up-to-date food choices and that this may influence the choice of healthier foods [25]. In terms of the participants’ abilities to properly rank products according to their nutritional quality, all labels showed some improvement compared to the GDA label. The Nutri-Score label showed the greatest improvements compared to the GDA label. There was also a significant improvement in the MTL label, followed by the HSR label, compared to the GDA label. These results among Greek consumers are in line with the findings of other countries such as: Belgium, Bulgaria, Denmark, France, Germany, Spain, Switzerland, the Netherlands, Italy and the United Kingdom based on the FOP-ICE study, where the Nutri-Score was observed as the best system to help participants determine the nutritional quality of products [20,26,27,28,29,30].

Another important finding of our study is about the perceptions of consumers, given that there was no familiarity with most of the labels we examined, except for the GDA in the Greek context. The perceptions of the consumers were generally favorable for all FoPLs. The Nutri-Score had the highest rating for the statements ‘I like this label’, ‘This label is easy to understand’, ‘I trust this label’ and, also, the Nutri-Score had the lowest rating for the statement ‘This label does not stand out’. It means the Nutri-Score was considered the most trusted, useful and easy to understand FoPL. The GDA label received the highest score in the question ‘It should be compulsory for this label to be shown on packaged food products’. This result is in line with the study conducted by Zenobia Talati and her colleagues [31].

In this context, it should be emphasized that, according to the international literature, interpretative FoPLs and color labels are superior in their recognition and interpretation and can lead to healthier and quicker food purchase [32,33]. The use of colors (from green to red) can be important for recognizing and understanding the label because they are recognized very quickly by the human eye (such labels include the Nutri-Score and the MTL). Monochrome labels such as the GDA, HSR & Warning Symbols are less noticeable to the human eye [29]. Corresponding conclusions emerge in our work, giving a very clear lead in the color and interpretive label of the Nutri-Score [34,35].

According to M. Cecchini and L. Warin, food labeling would increase the participation of people who choose a healthier food product by 17.95% [36]. In addition, according to a meta-analysis conducted by S. Shangguan et al. the use of food labeling can lead to a reduction in energy and fat consumption and an increase in vegetable consumption [37]. Studies indicate that consumers prefer food labels on the front of the package which are better perceived than the label on the back of the package [38]. A survey of low- and middle-income consumers in Mexico, shows that labels such as the Warning Symbols, Health Star Rating or Multiple Traffic Light can help consumers choose healthier foods compared to labels such as the Guideline Daily Amounts [25]. Moreover, in a study conducted on a sample of 814 participants in Morocco testing the Nutri-Score and four other nutritional information labels such as the Health Star Rating, Warning Symbols, Reference Intakes and Multiple Traffic Light, the results showed the superiority of the Nutri-Score, which was associated with the highest improvement in the ability to properly classify foods based on their nutritional quality compared to the GDA [39]. Finally, a study published in March 2021, investigated the effect of the Nutri-Score on food purchase intentions compared to foods that had the GDA, FoPL shape or no FoPL shape in three simulation studies in students (*N* = 1866), low-income people (*N* = 336) and people with cardiovascular diseases (*N* = 1180). In the simulation studies, the shopping cart contained more raw fruit and raw meat when the food had the Nutri-Score compared to the food basket when the food did not have a FoPL shape or had the GDA. The researchers found that when the Nutri-Score was present in food, consumers intended to buy more unprocessed products and reduce the purchase of highly processed foods, which is in line with dietary guidelines [20,40,41].

In addition, we should emphasize that, in the questions related to the choice of food, although it is again preceded by the Nutri-Score label, the second choice of Greek consumers is the GDA label. This was interpreted by our scientific team as the result of familiarity with this specific label because the packaged products that circulate in the Greek market bear this label. We should also note at this point that the GDA label helped modify choices in the right direction, but people can’t really understand the information that is provided [42].

Further, 53.4% of the participants stated that they saw the label during the survey, which leads to the conclusion that there is great room for improvement and education of the population for faster and easier observation and understanding of the food label [32].

### Limitation of Study

The labels of the study’s products were fictitious, and it was not possible for the participants to check the nutritional information on the back of the package. Another limitation is the stated preferences and their possible social desirability bias effects. In addition, due to the recruitment sample, we have an over representation of the highly educated population, as a result of the convenience of sampling. For the same reason, a large percentage of participants stated that they follow a healthy diet as well as that they have enough knowledge of nutrition. Finally, as the study was online, the price factor of the product is missing, which definitely affects the choices of consumers, especially in Greece after the economic crisis.

## 5. Conclusions

The findings of the study of Greek consumers provide data on the effectiveness of front-of-pack nutrition labels. The Nutri-Score seems to help Greek consumers better rank foods according to their nutritional value. We consider our study to be of crucial importance because policy makers will be facilitated in their procedure of making informed decisions regarding the food industry. This will lead to the reformulation of the ingredients of food products towards ensuring public health. In the future, more studies should be carried out to investigate the effect of labels only on low and medium socio-economic levels in Greece, as well as in adolescents, and, in the end, in a real buying environment. Public awareness campaigns are needed for Greek consumers to understand the label so that they can improve their food choices.

## Figures and Tables

**Figure 1 nutrients-14-00046-f001:**
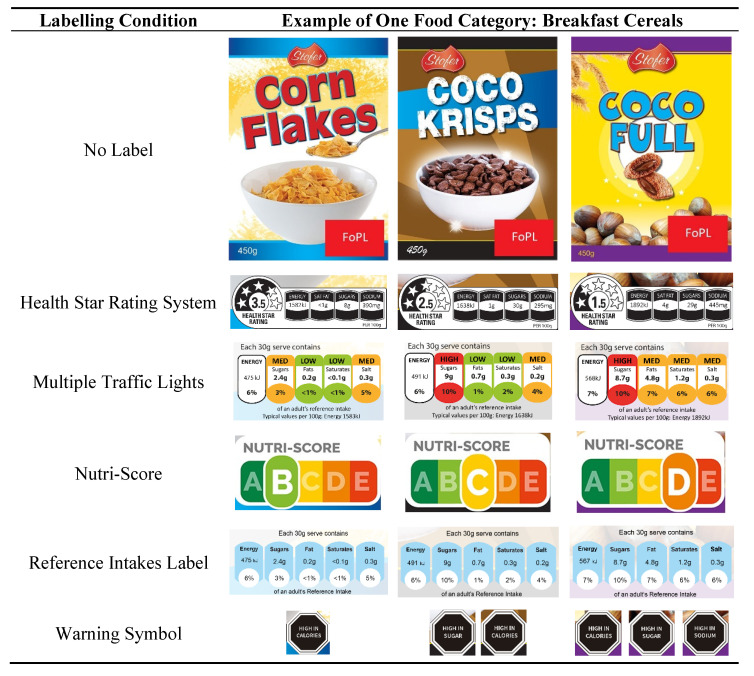
Example of a set of three products tested in the present study with the associated FoPLs (Front-of-pack labels).

**Figure 2 nutrients-14-00046-f002:**
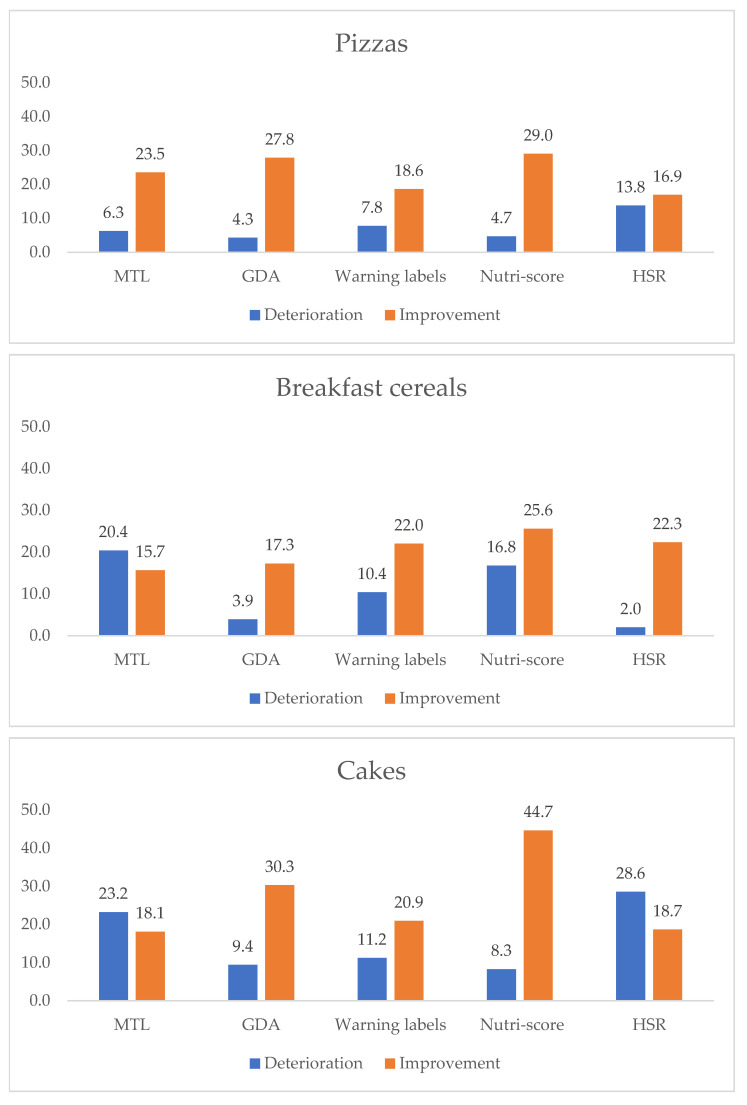
Percentage of participants who improved or deteriorated their food choices depending on the product category and its nutritional labeling in Greece.

**Figure 3 nutrients-14-00046-f003:**
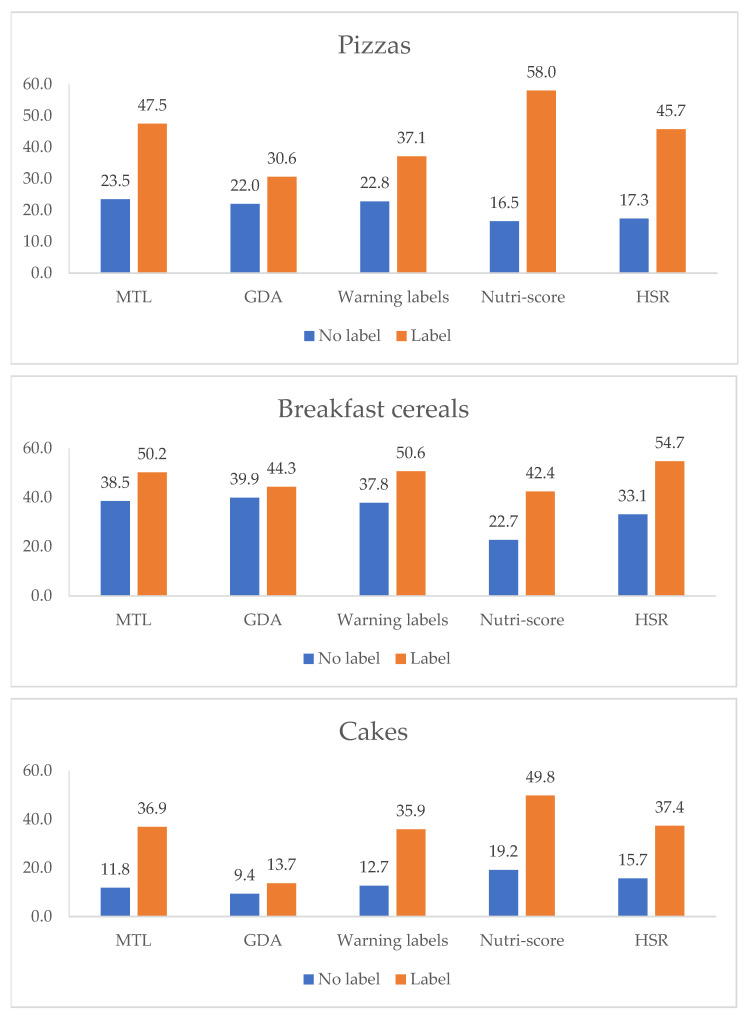
Percentage of correct answers in relation to the unlabeled and labeled situation depending on the type of food and FoPLs.

**Figure 4 nutrients-14-00046-f004:**
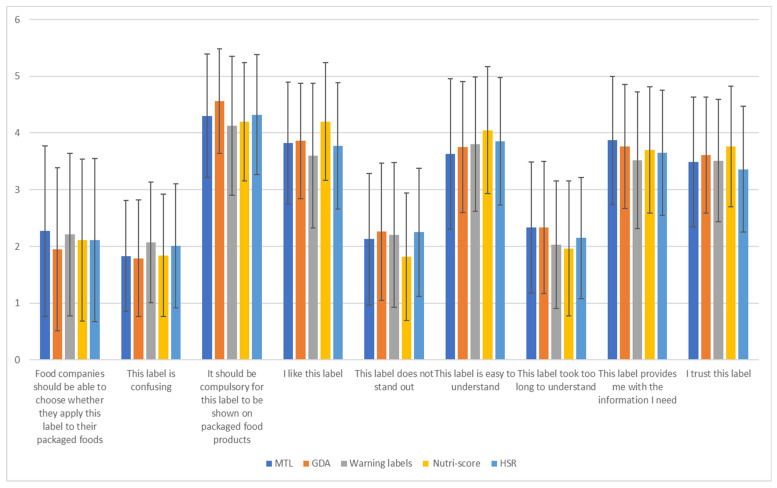
Perceptions of consumers.

**Figure 5 nutrients-14-00046-f005:**
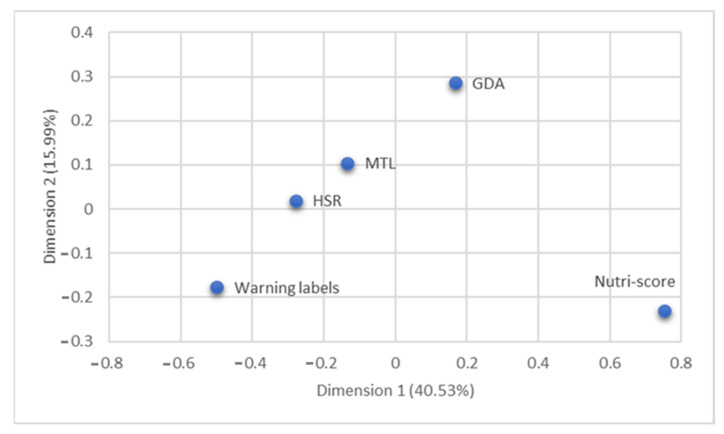
Placement map of each FoPL in two dimensions.

**Table 1 nutrients-14-00046-t001:** Individual characteristics of the study sample from Greece (*N* = 1278).

		*N*	%
Sex	Male	626	49.0%
	Female	652	51.0%
Age Groups	18–30	427	33.4%
	31–50	445	34.8%
	>50	406	31.8%
Income Groups	Low	431	33.7%
	Medium	464	36.3%
	High	383	30.0%
Educational level	Primary education	7	0.5%
	Secondary education	288	22.5%
	College certificate (IEK)	71	5.6%
	University bachelor degree	648	50.7%
	University postgraduate degree	264	20.7%
Responsible for grocery shopping	Yes	404	31.6%
	No	436	34.1%
	Share shopping with another household member	438	34.3%
Self-estimated diet quality	I follow a very unhealthy diet	20	1.6%
	I follow an unhealthy diet	165	12.9%
	I follow a healthy diet	1036	81.1%
	I follow a very healthy diet	57	4.5%
Self-estimated nutrition knowledge	I have no nutrition knowledge	5	0.4%
	I have little nutrition knowledge	500	39.1%
	I have adequate nutrition knowledge	634	49.6%
	I have very good nutrition knowledge	139	10.9%
Did you see the FoP label during the survey?	No	359	28.1%
	I am not sure	236	18.5%
	Yes	683	53.4%
Read the nutrition statement	No	214	16.7%
On the back of the product	Sometimes	535	41.9%
Packaging	Yes	529	41.4%
Participants who recalled seeing the FoPL they were exposed to	MTL	131	51.4%
	GDA	163	63.9%
	Warning Symbols	127	49.0%
	Nutri-Score	128	50.2%
	HSR	134	52.8%

**Table 2 nutrients-14-00046-t002:** Sample distribution across FoPLs.

FoPL					
		Frequency	Percent	Valid Percent	Cumulative Percent
Valid	MTL	255	19.9	19.9	20.0
	GDA	255	19.9	19.9	39.9
	Warning Symbols	259	20.3	20.3	60.2
	Nutri-Score	255	19.9	19.9	80.1
	HSR	254	19.9	19.9	100.0
	Total	1278	100.0	100.0	

**Table 3 nutrients-14-00046-t003:** Associations between front-of-pack label type and change in nutritional quality of food choices by food category (*n* = 1278) using Guideline Daily Amounts (GDA) label as the reference for the models.

Food Category	*N*	MTL		Warning Symbols		Nutri-Score		HSR	
		OR (95% CI)	*p*	OR (95% CI)	*p*	OR (95% CI)	*p*	OR (95% CI)	*p*
All categories	1265	0.71 [0.63–0.79]	<0.001	0.85 [0.76–0.96]	<0.01	1.07 [0.96–1.20]	0.2	0.75 [0.67–0.84]	<0.001
Pizza	1277	0.90 [0.69–1.16]	0.44	0.67 [0.51–0.87]	<0.01	1.06 [0.82–1.36]	0.6	0.54 [0.41–0.72]	<0.001
Cakes	1271	0.55 [0.45–0.67]	<0.001	0.79 [0.65–0.96]	0.02	1.17 [0.97–1.42]	0.09	0.54 [0.45–0.66]	<0.001
Breakfast Cereals	1270	0.59 [0.46–0.75]	<0.001	1.00 [0.79–1.28]	0.95	0.97 [0.76–1.24]	0.84	1.22 [0.95–1.55]	0.109

Multinomial logistic regression models were adjusted for sex, age, household monthly income level, education level, involvement in grocery shopping, nutrition knowledge, self-reported diet quality and whether the FOPL was noticed during participation in the study.

**Table 4 nutrients-14-00046-t004:** Associations between FoPL type and the ability to correctly rank products according to nutritional quality by food category (*N* = 1278) using Guideline Daily Amounts (GDA) label as the reference of the models.

Food Category	*N*	MTL		Warning Symbols		Nutri-Score		HSR	
		OR (95% CI)	*p*	OR (95% CI)	*p*	OR (95% CI)	*p*	OR (95% CI)	*p*
All categories	867	1.17 [1.05–1.29]	<0.01	1.09 [0.99–1.21]	0.076	1.36 [1.23–1.50]	<0.001	1.23 [1.11–1.36]	<0.001
Pizza	987	1.29 [1.05–1.58]	0.015	0.95 [0.77–1.17]	0.674	1.69 [1.39–2.05]	<0.001	1.34 [1.09–1.64]	<0.01
Cakes	954	1.49 [1.23–1.80]	<0.001	1.37 [1.13–1.65]	<0.01	1.53 [1.28–1.83]	<0.001	1.53 [1.26–1.84]	<0.001
Breakfast Cereals	988	0.96 [0.79–1.16]	0.68	1.08 [0.89–1.31]	0.429	1.27 [1.05–1.53]	0.013	1.17 [0.96–1.42]	0.107

Multinomial logistic regression models were adjusted for sex, age, household monthly income level, education level, involvement in grocery shopping, nutrition knowledge, self-reported diet quality and whether the FoPL was noticed during participation in the study.

**Table 5 nutrients-14-00046-t005:** Eigenvectors of each perceptions’ variable on the two dimensions from the principal component analysis.

	Eigenvectors	
	Dimension 1	Dimension 2
Food companies should be able to choose whether they apply this label to their packaged foods	−0.366	0.021
This label is confusing	−0.599	0.309
It should be compulsory for this label to be shown on packaged food products	0.594	0.480
I like this label	0.764	0.246
This label does not stand out	−0.500	0.658
This label is easy to understand	0.783	−0.053
This label took too long to understand	−0.500	0.673
This label provides me with the information I need	0.743	0.298
I trust this label	0.744	0.275

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
