# Peer review of "Online Consumer Survey Comparing Different Front-of-Pack Labels in Greece"

_nutrients, 2021, doi:10.3390/nu14010046_

Round 1
Reviewer 1 Report
The paper compares FOP nutrition labelling schemes in Greece. Although the paper is relevant for Greek policy makers, it has serious flaws that make it not appropriate for publication in its present form. Please find detailed comments below.
Introduction
The introductions lacks a detailed description of the different schemes. The authors should clearly describe their objectives and the type of information they provide. A summary of the available information should be included.
What is the novelty of the paper besides the application to the Greek context? Are Greek people already familiar with any of the schemes? A more detailed description of the Greek context is needed.
- Line 65 - The expression “very high” is not correct.
- The authors should avoid the use of single sentence paragraph.
Materials and Methods
The study has major methodological flaws and lack highly relevant information to judge the validity of the data.
- Recruitment seems highly biased and does not follow recommendations for best practice in online research. The authors used their personal networks and professional mail databases instead of relying on a recruitment method not involving personal contacts and institutional databases. In this sense, most of the participants have a University degree and report to follow a healthy diet, suggesting that the data lack external validity and do not reflect the behavior of the Greek population.
- No information is provided on how sample size as determined selected
- The authors did not compare the socio-demographic characteristics of the experimental groups to check that randomization was effective.
- Regarding the stimuli, the authors do not explain how the categories were selected and how their nutrient content was defined. In fact, no information about the nutritional composition of the different products included in the research is included in the paper, which makes it not possible to evaluate the validity and reliability of the results. In the same line, the authors do not indicate the nutrient profile model used for classifying products for each scheme.
- Another major concern is related to the fact that the different schemes show different information. For example, the MTL and the Warning labels classify the products differently. This suggests that the authors cannot compare the schemes, as they show different information. The authors should have used the same criteria for classifying nutrient content in low/medium/high. Similarly, the MTL and GDA shows information about salt, whereas the warning labels refer to sodium. This again makes the comparison among schemes not possible.
- The experimental procedures are not clearly described.
- The ranking task lacks ecological validity, as consumers would never rank products at the point of purchase. This task is highly biased and intentionally favors Nutriscore and the Health star rating, as they are designed to rank products. What is the purpose of using a task that only corresponds to the objective of one of the schemes. The authors should have consider other tasks that take into account the objective of the other schemes, for example asking whether products have high content of a specific nutrient.a
- Why did the authors rely on a within-subjects experimental design to compare the schemes. This could have introduced biases in the responses.
- Data analysis is not correctly described. The authors assigned arbitrary values to categorical responses and do not explain the type of model they used for analyzing the data.
- Why did the authors remove participants that would not buy any product? This is a highly relevant behavior in the context of the implementation of FOP nutrition labelling. In fact, it could largely change conclusions regarding the effectiveness of the schemes.
- The authors should have used Factor Analysis instead of PCA for analyzing their questionnaire
Results
The results are not clearly presented. The authors do not report the percentage of participants who were excluded from the analysis of the choice task. Statistical comparisons are not shown for many outcomes.
Discussion
The discussion is highly biased due to the major flaws of the experimental design. The authors do not acknowledge the limitations of their study and do not discuss the findings in the light of the large body of evidence that compares FOP nutrition labelling schemes.
Reviewer 2 Report
The paper investigates the understanding and perceptions of 1278 Greek consumers through an online survey in response to five different front-of-pack nutritional labels. The results from a regression model show the Nutri-score label to obtain the greatest improvements in healthy cakes, pizzas and breakfast cereals. Overall, the Nutri-score seems to help Greek consumers when ranking foods according to their nutritional value compared with the rest of food labeling used.
Overall, I think this is good work with very interesting results. Yet I have some specific comments, which might improve the quality of the manuscript. See my comments below; I hope it contributes to guiding you through your revision.
Pg. 1 line 35: Please define COSI
Pg. 2 lines 83-88: Is there any specific reason for choosing these FoP labels, besides that you are following the methodology and the questionnaire used in another study? Meaning, are there any statistics or data available showing the presence of the chosen food labels in the Greek market?
Pg. 3 lines 111-113: I thought the survey was directed to all types of food, as you do not mention any food categories neither in the introduction nor in the abstract of the manuscript. Please do that. Is there any specific reason for choosing these three food categories? Please mention that as well in the manuscript. Is it cereals or breakfast cereals Please homogenize this in the entire manuscript.
Pg. 3 lines 115-119: Please insert another separate paragraph to explain the survey content. How do you measure the self-rate diet quality, and how their nutritional knowledge? Include here the explanation of the food choice task, objective understanding, and perception, as well.
Pg. 3 lines 121-122: This part should be also mentioned in the introduction and also the abstract of the manuscript. It is here when you explain to the reader the reasons for choosing these three particular food categories to include in your study.
Pg. 3 lines 122-123: This reference does not correspond with the statement. Please check.
Pg. 5 lines 162-164: You should report this information as it is part of your results.
Pg. 5 line 173: How do you measure whether they noticed the FOP labels or not? Mention this in the section where you describe the survey and the content.
Pg. 5 lines 175-179: Move this part in the section where you explain the survey, as well.
Pg. 6 table 1: Did you show participants a specific figure or the FOP label when asking them whether they saw the FoP label during the survey?
Pg. 9 results of table 3: Perhaps I missed something but why do you use the GDA label as the baseline instead of using the unlabeled food package?
Pg. 13 line 344: Spain is mentioned twice.
Pg. 14 lines 396-400: Another limitation to mention is the stated preferences and their possible social desirability bias effects.
Pg. 14 – Conclusions: What are the implications of this study for policymakers (since you also mention “food policy” as a keyword? What kind of implications does this work offer for producers, manufacturers, and overall the food industry?
Reviewer 3 Report
I would like to congratulate the authors for the interesting work carried out, supported by a sample of 1278 participants. However, I believe that the job cannot be accepted for two main reasons.
The work lacks a theoretical delimitation that supports the relationships that are the object of analysis. It is necessary to identify works that have previously studied the different relationships that are presented. Likewise, the constructs under analysis must be delimited from a theoretical perspective.
Second, regarding the field work and method, the suitability of the scales used must be justified, explicitly indicating the source from which it was extracted. In the event that the scale has been prepared by the authors, explain the validation process followed. Likewise, the works that have been supported to establish the methodology followed in their work should be explained. The work does not indicate the source from which the scale is extracted, nor those works that serve as a support so that it can be considered an adequate method.
Minor elements are added to this.
The job title, I don't think the fact that an online survey was conducted should be highlighted. You must develop an attractive title that is able to arouse the interest of readers.
When using acronyms, present the full name the first time. For example, in the Abstract, it refers to the WHO without presenting its full name.
In section 2.1 participants, in the fourth paragraph (starting from the line) you repeat ideas that you have already presented in the section itself.
In the work it is stated on several occasions that they were “Greek participants”, was it somehow controlled that they were Greek participants or was the field work carried out in Greece?
81% of the sample affirms that they carry out a healthy diet. This bias must be taken into account in the presentation of the results, especially in their analysis. In total, 85.5% of the sample affirms that they follow a healthy or very healthy diet. On the contrary, only 60.5% of the sample affirms that they have adequate or very good nutritional knowledge. Do such values ​​have implications for the results of the work?
On line 135 there is a typo “[15-]”. Also, sometimes it has decimals and other times it doesn't. (line 110).
Round 2
Reviewer 1 Report
The authors have not addressed most of my concerns. They justify most of the flaws in the experimental design and data analysis based on their intention to compare results with those reported in previous studies. This is not a scientifically valid justification. In addition, the manuscript still does not provide an in-depth discussion of the published literature comparing FOP nutrition labelling schemes.
- The information provided in the manuscript lacks external validity as the authors recruited participants based on their personal networks and institutional databases.
- Results lack ecological validity as the main task used to compare the schemes does not reflect behavior in real life settings.
- The experimental groups are significantly different in educational level, as explained in the Response letter. This makes the comparison of the schemes not valid.
- The authors seemed to have used different criteria for classifying nutrient content in high/medium/low in the different schemes.
- Some of the data analysis are not correct, as indicated in my previous review.
In addition, key information is lacking to judge the validity of the experimental task. The authors still do not provide the nutritional composition of the products included in the task and do not explain the nutrient profile used for each of the schemes.
Reviewer 2 Report
I think that most of the comments have been addressed thus I want to congratulate the authors for that.
Author Response
Thank you for your comments and your effort to improve this manuscript.
Reviewer 3 Report
I am going to focus my review on the fundamental elements exposed in my first report.
It was indicated “The work lacks a theoretical delimitation that supports the relationships that are the object of analysis. It is necessary to identify works that have previously studied the different relationships that are presented. Likewise, the constructs under analysis must be delimited from a theoretical perspective”. The authors have included a paragraph supported by a single source, so they consider that the requested theretical delimitation has not been made.
In second place it was indicated “regarding the field work and method, the suitability of the scales used must be justified, explicitly indicating the source from which it was extracted. In the event that the scale has been prepared by the authors, explain the validation process followed. Likewise, the works that have been supported to establish the methodology followed in their work should be explained. The work does not indicate the source from which the scale is extracted, nor those works that serve as a support so that it can be considered an adequate method.” In this regard, I gave you two indications. In my opinion, validating a questionnaire involves more than indicating that it was translated into different languages. Likewise, the authors state that “Both the questionnaire and the methodology were based on 95 the study FOP-ICE (Front-Of-Pack International Comparative Experimental) study that 96 has been conducted so far in 18 countries [17]”. If the work [17] is consulted, it seems that additional variables have been incorporated and analyzes are carried out that do not appear in it. For this reason, it was necessary to indicate, precisely, the source from which the items included in the questionnaire were extracted at the same time as indicating those studies that had previously applied the methodology used. All this is necessary to establish the validity of the results.
